# Characterization of Antioxidant and α-Glucosidase Inhibitory Compounds of *Cratoxylum formosum* ssp. *pruniflorum* and Optimization of Extraction Condition

**DOI:** 10.3390/antiox12020511

**Published:** 2023-02-17

**Authors:** Heewon An, Le Nguyen Thanh, Le Quoc Khanh, Se Hwan Ryu, Solip Lee, Sang Won Yeon, Hak Hyun Lee, Ayman Turk, Ki Yong Lee, Bang Yeon Hwang, Mi Kyeong Lee

**Affiliations:** 1College of Pharmacy, Chungbuk National University, Cheongju 28160, Republic of Korea; 2Department of Medicinal Chemistry Technology, Institute of Marine Biochemistry, Vietnam Academy of Science & Technology (VAST), Hanoi 10000, Vietnam; 3Graduate University of Science and Technology, VAST, Hanoi 10000, Vietnam; 4Hatinh Pharmaceutical Company (HADIPHAR), Ha Tinh 45000, Vietnam; 5College of Pharmacy, Korea University, Sejong 47236, Republic of Korea

**Keywords:** *Cratoxylum formosum* ssp. *pruniflorum*, xanthone, benzophenone, antioxidant, α-glucosidase, response surface methodology

## Abstract

*Cratoxylum formosum* ssp. *pruniflorum* (Kurz.) Gogel (Guttiferae), called kuding tea, is widely distributed in Southeast Asia. In this study, the constituents and biological activity of *C. formosum* ssp. *pruniflorum* were investigated. Extract of its leaves, roots and stems showed antioxidant and α-glucosidase inhibitory activity. Interestingly, comparison of the metabolite profiles of leaves, roots and stems of *C. formosum* ssp. *pruniflorum* by LC-MS analysis showed a great difference between the roots and leaves, whereas the roots and stems were quite similar. Purification of the roots and leaves of *C. formosum* ssp. *pruniflorum* through various chromatographic techniques resulted in the isolation of 25 compounds. The structures of isolated compounds were elucidated on the basis of spectroscopic analysis as 18 xanthones, 5 flavonoids, a benzophenone and a phenolic compound. Among them, a xanthone (**16**) and a benzophenone (**19**) were first reported from nature. Evaluation of biological activity revealed that xanthones had a potent α-glucosidase inhibitory activity, while flavonoids were responsible for the antioxidant activity. To maximize the biological activity, yield and total phenolic content of *C. formosum* ssp. *pruniflorum*, extraction conditions such as extraction solvent, time and temperature were optimized using response surface methodology with Box–Behnken Design (BBD). Regression analysis showed a good fit of the experimental data, and the optimal condition was obtained as MeOH concentration in EtOAc, 88.1%; extraction time, 6.02 h; and extraction temperature 60.0 °C. α-Glucosidase inhibitory activity, yield and total phenolic content under the optimal condition were found to be 72.2% inhibition, 10.3% and 163.9 mg GAE/g extract, respectively. These results provide useful information about *C. formosum* ssp. *pruniflorum* as functional foods for oxidative stress–related metabolic diseases.

## 1. Introduction

Diabetes is one of the most common metabolic diseases worldwide. According to the International Diabetes Federation, 463 million adults had diabetes worldwide as of 2019, with these numbers increasing to 578 million by 2030 and 700 million by 2045. In diabetes, the increased blood glucose level leads to release of glucose into the urine. Diabetes is caused by a malfunction of carbohydrate metabolism due to insufficient or abnormal insulin function. Sustained hyperglycemia progresses to various diabetes complications such as cardiovascular diseases, nephropathy, neuropathy and retinopathy [1,2,3]. 

Oxidative stress is caused by the excessive production of reactive oxygen species (ROS), which results from an increase in free radical production and/or a decrease in endogenous antioxidant defenses. Persistent oxidative stress by excessive production of ROS eventually leads to diverse severe diseases such as cancer, inflammation and metabolic diseases. The ROS production increases in diabetes, which exacerbates the inflammatory response and causes complications of diabetes [4,5,6]. 

Research to develop therapeutic agents for diabetes is being actively conducted in various ways [7,8]. Suppression of the increase in blood sugar is the primary therapeutic target for diabetes. Ingested carbohydrates are broken down into single monosaccharides to be absorbed, and α-glucosidase plays an important role in this process. Therefore, α-glucosidase inhibitors are suggested to retard the absorption of carbohydrates in the small intestine and further reduce postprandial glucose. Several α-glucosidase inhibitors such as acarbose and voglibose are used for the treatment of carbohydrate-mediated diseases [9,10]. Antioxidant action is also used as a therapeutic strategy to suppress the onset and complications of diabetes [11,12].

Natural products contain many substances with various activities, so they are important materials for disease treatment. Natural products with α-glucosidase inhibitory activity have been considered important targets for the treatment and prevention of diabetes by controlling blood glucose [13,14]. The antioxidant effect of natural products is widely known and has long been used for the prevention and treatment of diseases. Among various types of compounds, polyphenols are rich in plants and considered beneficial for oxidative stress and metabolic diseases [15,16]. 

*Cratoxylum formosum* ssp. *pruniflorum* (Kurz.) Gogel (Guttiferae) is widely distributed in Southeast Asia. It is also called kuding tea and has been used routinely in traditional foods and remedies for the treatment of metabolic diseases, inflammation, fever, coughs and diarrhea [17,18]. Investigations have revealed xanthones, flavonoid and terpenes as constituents [19,20,21,22]. Anti-cancer potentials of this plant have been suggested by many researchers [23,24,25,26]. In addition, various biological activities, including neuroprotective, anti-inflammatory, and antibacterial effects, were also reported [27,28,29].

In this study, the constituents and biological activity of the leaves, roots and stems of *C. formosum* ssp. *pruniflorum* were investigated. The metabolic profiles and biological activities of the leaves, roots and stems of *C. formosum* ssp. *pruniflorum* were compared by LC-MS/MS analysis. The constituents were purified and characterized. In addition, its antioxidant and α-glucosidase inhibitory activity were evaluated. For efficient use, the optimal extraction condition that maximizes efficacy and yield was also established using response surface analysis. 

## 2. Materials and Methods

### 2.1. Plant Material

The leaves, roots and stems of *C. formosum* ssp. *Pruniflorum*,were collected from trees of 3–4 m height at Huongkhe District at Hatinh Province (GPS: 18°24′30.3″ N 105°25′54.1″ E, 34 m), Vietnam, by Ha Tinh Pharmaceutical Company (HADIPHAR) in August 2019. After identification by Prof. Tran The Bach at Institute of Ecology and Biological Resources—Vietnam Academy of Science and Technology, voucher specimens (CBNU2019-CFL, CFR and CFS for leaves, roots and stems, respectively) were deposited in a specimen room of the herbarium of the College of Pharmacy Chungbuk National University.

### 2.2. General Experimental Procedure

A Bruker DRX 400 or 500 MHz spectrometer (Bruker-Biospin, Karlsruhe, Germany) was used for the analysis of NMR signals using methanol-*d_4_* as a solvent. The UV and IR spectra were obtained using Jasco UV-550 (JASCO, Tokyo, Japan) and Perkin–Elmer model LE599 (Perkin–Elmer, Waltham, MA, USA) spectrometers, respectively. ESIMS and HRESI-TOF-MS data were obtained with LCQ Fleet and maXis 4G mass spectrometers (Bruker Daltonics, Bremen, Germany), respectively. Semi-preparative HPLC (Waters, Milford, MA, USA) was performed using a Waters 515 HPLC pump with a 996-photodiode array detector, and Waters Empower software using a Gemini-NX ODS-column (150 × 10.0 mm and 150 × 21.2 mm). Column chromatography procedures were performed using silica gel (200–400 mesh, Fisher Scientific, Waltham, MA, USA) and Sephadex LH-20 (25–100 µm, Pharmacia Fine Chemical Industries Co., Uppsala, Sweden). Thin-layer chromatography (TLC) was performed using aluminum plates precoated with Kieselgel 60 F_254_ (0.25 mm, Merck, Darmstadt, Germany).

### 2.3. Analysis of Chemical Profile Using LC-MS/MS 

For the LC-HRMS/MS study, an Orbitrap Exploris 120 mass spectrometer was linked to a Vanquish UHPLC and diode array detector. The extracts of the leaves, roots and stems of *C. formosum* ssp. *pruniflorum* (0.5 mg/mL) were analyzed by YMC-Triart C18 column (100 × 2.1 mm, 1.9 μm), using a gradient system (H_2_O with 0.1% formic acid—CH_3_CN with 0.1% formic acid, 90:10 to 0:100) with a flow rate of 0.3 mL/min. The column oven was preheated to 30 °C, and the injection volume of samples was set at 5 μL. Orbitrap mass analyzer resolution was set at 60,000 for the whole MS scan and 15,000 for the data-dependent MS^n^ scan, and mass detection was performed in the m/z range of 200–2000. Spray voltage of 3.5 kV, vaporizer temperature of 275 °C, ion transfer tube temperature of 320 °C, sheath gas flow rate of 6.4 L/min, aux gas flow rate of 12 L/min, and sweep gas flow rate of 2.2 L/min were the ion source characteristics for HESI. Ion collisions in the Orbitrap detector occurred at a normalized higher-energy collision dissociation (HCD) energy of 30%. The four most intense ions’ MS^2^ spectra were acquired using MS/MS fragmentation with the data-dependent MS^n^ mode, and a dynamic exclusion filter was used to prevent further fragmentation of the ions within 2.5 s after getting the MS^2^ spectrum. 

### 2.4. Measurement of Antioxidant and α-Glucosidase Activity

The inhibitory effect on α-glucosidase was measured using α-glucosidase from *Saccharomyces cerevisiae* (EC 3.2.1.20) [30]. A test sample was mixed with 80 μL enzyme buffer and 10 μL α-glucosidase and incubated for 15 min at 37 °C. Then, after the addition of 10 μL *p*-nitrophenyl α-D-glucopyranoside solution for enzyme reaction, the amount of *p*-nitrophenol that was cleaved by the enzyme was determined by measuring the absorbance at 405 nm in a 96-well microplate reader. Acarbose was used as a positive control. The antioxidant activity was evaluated by measuring the DPPH radical scavenging activity using ascorbic acid as a positive control [30]. 

### 2.5. Quantitation of Phenolic and Flavonoid Contents

The leaves, roots and stems of *C. formosum* ssp. *pruniflorum* were extracted respectively with 80% MeOH. The total amounts of phenolic and flavonoid contents of each extract were quantitated using Folin–Ciocalteu assay and aluminum chloride colorimetirc assay, respectively [31,32]. 

### 2.6. Extraction and Isolation

For the purification of compounds, the dried powder of *C. formosum* ssp. *pruniflorum* root (87.0 g) was extracted with 80% MeOH (1 L × 2) at room temperature. The MeOH extract (4.8 g) was suspended in H_2_O and partitioned successively with CH_2_Cl_2,_ EtOAc and *n*-BuOH.

The CH_2_Cl_2_ fraction (CPRC, 1.3 g) was chromatographed on Sephadex LH-20 eluted with a mixture of *n*-hexane-CH_2_Cl_2_-MeOH (5:5:1) to obtain 17 subfractions (CPRC1-C17). Subfraction CPRC4 was subjected to semi-preparative HPLC eluted with acetonitrile-H_2_O (80:20) to yield compounds **3**, **6** and **13**. Semi-preparative HPLC (acetonitrile-H_2_O, 80:20) of CPRC5 and CPRC17 gave compounds **10** and **14**, and compounds **4**, **7**, **8** and **15**, respectively. Compounds **9** and **16** were isolated from CPRC11 and compounds **1** and **12** from CPRC8, respectively, by semi-preparative HPLC eluted with acetonitrile-H_2_O (55:45). Compounds **2**, **5** and **11** were purified from CPRC9, CPRC15 and CPRC7, respectively, by semi-preparative HPLC (acetonitrile-H_2_O, 40:60). 

The dried powder of *C. formosum* ssp. *pruniflorum* leaves (73.1 g) was extracted with 80% MeOH (1 L × 2) at room temperature. The MeOH extract (13.9 g) was suspended in H_2_O and partitioned successively with CH_2_Cl_2,_ EtOAc and *n*-BuOH.

The EtOAc fraction (CPLE, 1.6 g) was chromatographed on Sephadex LH-20 eluted with a mixture of CH_2_Cl_2_-MeOH (9:1) to obtain eight subfractions (CPLE1-E8). Compound **18** was isolated from CPRE6 by recrystallization. CPLE3 was subjected to Sephadex LH-20 chromatography eluted with CH_2_Cl_2_-MeOH (9:1) to yield seven subfractions (CPLE3A-G). Compounds **17** and **24** were purified from CPLE3E and compound **19** from CPLE3G, respectively, by semi-preparative HPLC eluted with acetonitrile-H_2_O (18:82). Semi-preparative HPLC (acetonitrile-H_2_O, 18:82) of CPLE4 and CPLE5 using acetonitrile-H_2_O (15:85) as eluent gives compound **20** and compounds **21**, **22**, **23** and **25**, respectively.

#### 2.6.1. Pruniflonone A (**16**)

Brown amorphous powder; ${\left[\alpha \right]}_{D}^{25}$ + 3.5 (*c* 0.01, MeOH); FT-IR *ν*_max_ 3680, 1558 cm^−1^; ^1^H-NMR (methanol-*d*_4_, 400 MHz) *δ*_H_ 7.49 (1H, d, *J* = 2.9 Hz, H-8), 7.38 (1H, d, *J* = 9.0 Hz, H-5), 7.24 (1H, dd, *J* = 2.9, 9.0 Hz, H-7), 5.54 (1H, brs, H-6″), 5.53 (1H, m, H-5″), 5.27 (1H, t, *J* = 7.1 Hz, H-2″), 5.20 (1H, t, *J* = 7.1 Hz, H-2′), 3.56 (2H, d, *J* = 7.1 Hz, H-1″), 3.39 (2H, d, *J* = 7.1 Hz, H-1′), 2.67 (2H, m, H-4″), 1.89 (3H, s, CH_3_-10″), 1.80 (3H, s, CH_3_-4′), 1.67 (3H, s, CH_3_-5′), 1.18 (6H, s, CH_3_-8″, 9″); ^13^C-NMR (methanol-*d*_4_, 100 MHz) *δ*_C_ 180.8 (C-9), 157.9 (C-2), 153.8 (C-5a), 153.1 (C-3), 149.8 (C-6), 147.9 (C-4a), 139.1 (C-6″), 133.8 (C-3″), 131.2 (C-3′), 124.5 (C-5″), 123.8 (C-7), 122.9 (C-2″), 121.9 (C-2′), 120.5 (C-8a), 120.5 (C-9a), 118.4 (C-5), 110.2 (C-1), 107.9 (C-8), 105.9 (C-4), 69.7 (C-7″), 42.0 (C-4″), 28.5 (C-8″, C-9″), 24.6 (C-5′), 21.3 (C-1″), 21.0 (C-1′), 16.6 (C-4′), 15.2 (C-10″); HRESI-TOF-MS m/z 463.2124 [M-H]^−^ (calcd. 463.2126) (Appendix A).

#### 2.6.2. Pruniflonone B (**19**)

Brown syrup; ${\left[\alpha \right]}_{D}^{25}$ − 45.5 (*c* 0.01, MeOH); FT-IR *ν*_max_ 3709, 1056 cm^−1^; ^1^H-NMR (methanol-*d*_4_, 400 MHz) *δ*_H_ 7.69 (1H, m, H-2′), 7.52 (1H, m, H-4′), 6.21 (1H, d, *J* = 1.9 Hz, H-2), 6.07 (1H, d, *J* = 1.9 Hz, H-6), 7.41 (1H, m, H-3′), 7.41 (1H, m, H-5′), 7.69 (1H, m, H-6′), 4.81 (1H, d, *J* = 7.7 Hz, H-1″) and ^13^C-NMR (methanol-*d*_4_, 100 MHz), *δ*_C_ 109.1 (C-1), 95.8 (C-2), 159.7 (C-3), 164.0 (C-4), 162.0 (C-5), 98.1 (C-6)], 141.6 (C-1′), 130.1 (C-2′), 129.0 (C-3′), 133.2 (C-4′), 129.0 (C-5′), 130.1 (C-6′)], 199.5 (C-7), 101.9 (C-1″), 73.3 (C-2″), 76.0 (C-3″), 70.6 (C-4″), 74.3 (C-5″), 63.5 (C-6″)]; HRESI-TOF-MS m/z 391.1034 [M-H]^−^ (calcd. 391.1035) (Appendix A).

### 2.7. Response Surface Methodology

A Box–Behnken design (BBD) with three variables such as extraction solvent (X_1_), extraction time (X_2_) and extraction temperature (X_3_) was chosen, with the three variables serving as independent variables, and α-glucosidase inhibitory effects together with yield and total phenolic content were determined as the dependent responses. Regression analysis was performed according to the experimental data; the mathematical model can be explained by the following equation: Y is the response, β0 is the constant coefficient, βi are the linear coefficients, βii are the quadratic coefficients and βij are the interaction coefficients. The statistical significance of the coefficients in the regression equation was checked by analysis of variance (ANOVA). The fitness of the polynomial model equation to the responses was evaluated with the coefficients of R^2^.

## 3. Results and Discussion

### 3.1. Comparison of Different Parts of C. formosum ssp. pruniflorum

Plant components are synthesized through plant-specific biosynthetic pathways, so there are similarities throughout the plant. However, if you subdivide it a little more, it shows some differences in constituents for each part of the plant, which leads to a difference in efficacy [32,33,34]. Therefore, we first compared the antioxidant and anti-diabetic efficacy of the parts of this plant, such as leaves, stems and roots. Since xanthone and flavonoid components have been known as major components of this plant, the contents in each part of the plant were also compared.

As shown in Table 1, all the parts of this plant, including the leaves, roots, and stems, showed antioxidant and α-glucosidase inhibitory effects. However, there were differences in the efficacy. The antioxidant effect was observed most strongly in the leaves, and the roots and stems also showed the efficacy. However, the roots showed the most excellent α-glucosidase inhibitory efficacy with an IC_50_ value of 2.0 μg/mL, followed by the leaves with an IC_50_ value of 3.9 μg/mL, but relatively weak efficacy in the case of the stem. As a result of comparing the contents of components, both flavonoid and phenol contents were highest in leaves. In particular, in the case of leaves, the content of flavonoid was relatively high, whereas phenolic compounds were observed to be high in roots and stems. 

We further compared the chemical profiles of each part of *C. formosum* ssp. *pruniflorum.* As shown in Figure 1, the MS/MS chromatogram of leaves of *C. formosum* ssp. *pruniflorum* was quite different from that of roots and stems. Peak analysis by LC-MS/MS showed that mangiferin and quercetin-3-*O*-glucopyranoside were the major constituents of leaves, whereas α-mangostin, 7-geranyloxy-1,3-dihydroxyxanthone, and cochinchinone A were the major constituents of roots and stems (Table 2). The chemical patterns of the roots and stems were quite similar, but the components of the roots were more diverse than those of the stems and showed a higher content. Therefore, roots and leaves were selected for further purification of compounds.

### 3.2. Isolation and Characterization of the Constituents of C. formosum ssp. pruniflorum

Using various chromatography methods, 16 (**1–16**) and 9 (**17–25**) compounds were isolated from the roots and leaves of this plant, respectively. The structures of the isolated compounds were identified using spectroscopic methods as 2 new compounds (**16** and **19**) together with 23 known compounds. 

#### 3.2.1. Structure Elucidation of New Compounds

Compound **16** was isolated as a light brown amorphous powder. The molecular formula of **16** was determined as C_28_H_31_O_6_ from the HRESIMS (m/z 463.2124 [M-H]^−^, calcd. 463.2126) and the ^13^C NMR data. The characteristic UV absorption at 233, 266 and 317 nm suggested compound **16** as a xanthone skeleton [35]. The presence of the xanthone skeleton was confirmed from the 12 aromatic signals at [*δ*_C_ 110.2 (C-1), 157.9 (C-2), 153.1 (C-3), 105.9 (C-4), 147.9 (C-4a), 118.4 (C-5), 153.8 (C-5a), 149.8 (C-6), 123.8 (C-7), 107.9 (C-8), 120.5 (C-8a), 120.5 (C-9a)] together with a carbonyl carbon at *δ*_C_ 180.8 (C-9) in the ^13^C NMR spectrum and the signals for a 1,3,4-trisubstituted benzene ring at [*δ*_H_ 7.38 (1H, d, *J* = 9.0 Hz, H-5), 7.24 (1H, dd, *J* = 2.9, 9.0 Hz, H-7), 7.49 (1H, d, *J* = 2.9 Hz, H-8)] in the ^1^H NMR spectrum, which was also supported by the HSQC spectrum. The presence of a prenyl group was deduced by the signals at [*δ*_H_ 3.39 (2H, d, *J* = 7.1 Hz, H-1′), 5.20 (1H, t, *J* = 7.1 Hz, H-2′), 1.80 (3H, s, CH_3_-4′), 1.67 (3H, s, CH_3_-5′); *δ*_C_ 21.0 (C-1′), 121.9 (C-2′), 131.2 (C-3′), 16.6 (C-4′), 24.6 (C-5′)]. Additionally, the signals at [*δ*_H_ 3.56 (2H, d, *J* = 7.1 Hz, H-1″), 5.27 (1H, t, *J* = 7.1 Hz, H-2″), 2.67 (2H, m, H-4″), 5.53 (1H, m, H-5″), 5.54 (1H, brs, H-6″), 1.18 (6H, s, CH_3_-8″, 9″), 1.89 (3H, s, CH_3_-10″); *δ*_C_ 21.3 (C-1″), 122.9 (C-2″), 133.8 (C-3″), 42.0 (C-4″), 124.5 (C-5″), 139.1 (C-6″), 69.7 (C-7″), 28.5 (C-8″, C-9″), 15.2 (C-10″)] suggested the presence of a geranyl group. The presence of hydroxy group in the geranyl group was suggested by the oxymethine carbon at *δ*_C_ 69.7 (C-7″) and two methyl signals at *δ*_H_ 1.18 (6H, s, CH_3_-8″, CH_3_-9″), which was confirmed by the HMBC correlations from CH_3_-8″, 9″ to C-7″. Therefore, compound **16** was suggested to be a xanthone derivative with prenyl and hydroxygeranyl moieties. The positions of the prenyl and hydroxygeranyl moieties were deduced to C-1 and C-4, respectively, by the correlation from H-1′ to C-1 and from H-1″ to C-4 in the HMBC spectrum. On the basis of the obtained data, compound **16** was determined as shown and named pruniflonone A. 

Compound **19** was purified as brown syrup with the molecular of C_19_H_20_O_9_ by HRESI-TOF-MS analysis (m/z 391.1034, calcd. for C_19_H_19_O_9_^−^, 391.1035) and ^13^C NMR data. The ^1^H and ^13^C NMR spectra revealed the signals of a tetrasubstituted aromatic ring at [*δ*_H_ 6.21 (1H, d, *J* = 1.9 Hz, H-2), 6.07 (1H, d, *J* = 1.9 Hz, H-6); *δ*_C_ 109.1 (C-1), 95.8 (C-2), 159.7 (C-3), 164.0 (C-4), 162.0 (C-5), 98.1 (C-6)], a monosubstituted aromatic ring at [*δ*_H_ 7.69 (2H, m, H-2′, 6′), 7.41 (1H, m, H-3′, 5′), 7.52 (1H, m, H-4′); *δ*_C_ 141.6 (C-1′), 130.1 (C-2′, 6′), 129.0 (C-3′, 5′), 133.2 (C-4′)] and a carbonyl carbon at *δ*_C_ 199.5 (C-7). The presence of a glucosyl moiety was also confirmed by an anomeric proton at *δ*_H_ 4.81 (1H, d, *J* = 7.7 Hz, H-1″) together with the glucosyl carbon signals at [*δ*_C_ 101.9 (C-1″), 73.3 (C-2″), 76.0 (C-3″), 70.6 (C-4″), 74.3 (C-5″), 63.5 (C-6″)]. The HMBC correlations from H-2/6 and H-2′/6′ to C-7 (C=O) suggested the presence of a benzophenone skeleton. The position of the glucose was determined to be located at C-3 on the basis of HMBC correlation from the anomeric proton (H-1″) to C-3. Based on these data, compound **19** was determined as shown in Figure 2 and named pruniflonone B.

#### 3.2.2. Identification of Known Compounds

The known compounds were identified as 17 xanthones—cochinchinone F (**1**), 1,3,7-cratosumatranone D (**2**), isocudraniaxanthone B (**3**), viellardixanthone B (**4**), diisoprenylxanthone (**5**), *γ*-mangostin (**6**), *α*-mangostin (**7**), *β*-mangostin (**8**), garcinone C (**9**), garcinone D (**10**), 11-hydroxy-1-garciniacowones E (**11**), isomangostin (**12**), garcinone B (**13**), trihydroxy-2,4-7-geranyloxy-1,3-dihydroxyxanthone (**14**), cochinchinone A (**15**), caloxanthone E (**17**) and mangiferin (**18**); a phenolic compound—protocatechuic acid (**20**); and 5 flavonoids—epicatechin (**21**), quercetin-3-*O*-glucopyranoside (**22**), isorhamnetin-3-*O*-glucoside (**23**), gujaverin (**24**) and quercetin-3-*O*-*α*-L-rhamnoside (**25**) via analysis of their physical data and comparison with values in the literature [36,37,38,39,40,41,42,43,44,45,46,47,48,49,50,51,52,53,54,55,56]. 

### 3.3. Evaluation of Antioxidant and α-Glucosidase Inhibitory Activity

The biological activity of the isolated compounds were evaluated by measuring the DPPH radical scavenging and α-glucosidase inhibitory activity. As described above, compounds isolated from *C. formosum* ssp. *pruniflorum* in this study are aromatic compounds and can be subdivided according to the compound skeleton as follows: xanthones (**1–18**), a benzophenone (**19**), a simple phenolic (**20**) and flavonoids (**21–25**). These isolated compounds showed good antioxidant and α-glucosidase inhibitory activity but differential efficacy depending on the structures (Figure 3). 

Xanthones are more effective in the inhibition of α-glucosidase activity, whereas flavonoids are effective in antioxidant activity. Xanthones inhibited α-glucosidase activity with IC_50_ values of <50 μM. However, the addition of a hydroxyl group to prenyl or geranyl groups reduced the efficacy, as observed in **1** and **9**. The addition of a sugar moiety also showed negative effects on α-glucosidase inhibition. In the case of antioxidant activity, xanthones **3**, **6**, **9**, **17** and **18** showed more than 50% DPPH radical scavenging activity at 50 μM. Considering the structure, dihydroxy groups are important for the antioxidant activity of xanthones. In the case of flavonoids, flavonoids except compound **23** showed good antioxidant activity. Similar to xanthones, flavonoids with dihydroxy groups exerted antioxidant activity. Related to the α-glucosidase inhibitory activity of flavonoids of *C. formosum* ssp. *pruniflorum*, compound **21** without any sugar moieties showed good inhibition. However, benzophenone (**19**) exerted a weak effect on both antioxidant and α-glucosidase inhibition. 

As described in Table 1, the extract of *C. formosum* ssp. *pruniflorum* exhibited α-glucosidase inhibitory and antioxidant activity. It contains xanthones, flavonoids and benzophenone, and most of them showed α-glucosidase inhibitory and/or antioxidant activity (Figure 3). In the case of the newly reported compounds in this study, compound **19** showed antioxidant efficacy, but compound **16**, unfortunately, had weak efficacy. Conclusively, although the efficacies of compounds were quite different in each compound, xanthones and flavonoids were suggested to contribute to the antioxidant and α-glucosidase inhibitory potentials of *C. formosum* ssp. *pruniflorum*.

Differences were also observed depending on plant parts. For the α-glucosidase inhibition, the root extract showed the best activity, whereas the leaf extract showed the strongest antioxidant activity. Investigation of the constituents showed that the roots contained xanthones as major components and the leaves had flavonoids, which were consistent with the HRESI-MS/MS chromatogram (Figure 1). Measurement of biological activities of isolated compounds suggested that xanthone had α-glucosidase inhibitory potential, whereas flavonoids were more effective in antioxidant activity, which supported differential efficacy of the extract for each part. 

Taken together, these results suggested the components and efficacy of *C. formosum* ssp. *pruniflorum*, which are differential depending on each part and can be used for the development of a marker component of each part. 

### 3.4. Optimization of Extraction Conditions Using Response Surface Metholodogy

The roots of *C. formosum* ssp. *pruniflorum* showed strong α-glucosidase inhibitory effects, and xanthones were assigned as active compounds. The content of active constituents in extract is highly affected by extraction conditions such as extraction solvent, extraction time and extraction temperature, which resulted in the difference in their biological activity [57,58]. Therefore, we further optimized the extraction conditions for maximum α-glucosidase inhibitory effects. Response surface methodology (RSM) is a statistical tool that takes several factors into account simultaneously using rationally designed experiments. The optimal condition can be derived effectively, especially in the case of several variables [59,60]. Therefore, RSM using a Box–Behnken design (BBD) was chosen for the optimization of extraction conditions of *C. formosum* ssp. *pruniflorum* for maximum efficiency.

Three variables such as extraction solvent (X_1_), extraction time (X_2_) and extraction temperature (X_3_) were chosen as independent variables, and the range of each variable was determined in the preliminary study. α-Glucosidase inhibitory effects together with yield and total phenolic content were determined as the dependent responses. The variables were coded at three levels (−1, 0 and 1), and the complete design consisted of 15 experimental points including three replications of the center points whose variables were all coded as zero (Table 3). Multiple regression analysis of the experiment data yielded the following second-order polynomial regression equation: 

α-Glucosidase inhibition = 67.83 + 20.47X_1_ + 4.39X_2_ − 2.81X_3_ − 19.25X_1_^2^ + 0.43X_2_^2^ − 2.80X_3_^2^ − 2.40X_1_X_2_ + 2.43X_1_X_3_ + 2.61X_2_X_3._

Yield = 7.83 + 2.77X_1_ − 0.02X_2_ + 0.23X_3_ − 0.87X_1_^2^ − 0.33X_2_^2^ + 0.26X_3_^2^ + 0.54X_1_X_2_ + 0.45X_1_X_3_ − 0.10X_2_X_3._

Total phenolic content = 163.28 + 33.00X_1_ − 2.15X_2_ + 3.88X_3_ − 46.44X_1_^2^ + 2.38X_2_^2^ − 6.11X_3_^2^ + 1.50X_1_X_2_ + 6.38X_1_X_3_ − 0.95X_2_X_3._

The values of the coefficient determination (R2) and the adjusted coefficient determination (adj. R2) of the predicted model in this response suggested that the regression equation can explain the observed value to a high degree. Insignificant *p*-values of lack of fit (>0.05) for three responses also indicated the adaptability of this analysis (Table 4). 

Among extraction variables, the linear term (X_1_) of MeOH concentration showed the most significant effect on all three responses. Relationships between the two variables in each response were also shown in a three-dimensional response surface (Figure 4). Consistent with multiple regression analysis, extraction solvent showed the strongest effect on yield, phenolic content and α-glucosidase inhibition (Figure 4A,D,G). Yield was increased with increasing MeOH concentration, but phenolic content and α-glucosidase inhibition were decreased with a continuing increase in MeOH concentration. On fixed temperature at 40 °C, yield was also affected by extraction time (Figure 4B), whereas total phenolic content was affected by extraction temperature when extracted with the mixture of MeOH-EtOAc (1:1) (Figure 4E). However, compared with extraction solvent, α-glucosidase inhibition showed slight changes as extraction time and temperature changed.

Based on these results, the extraction condition for maximum yield, α-glucosidase inhibitory effects and total phenolic content was optimized. The extract prepared using the optimized extraction condition was found to exert 73.9% α-glucosidase inhibitory effects at 1 μg/mL with a yield of 10.9% and a total phenolic content of 163.9 mg GAE/g extract (Table 5). The total phenolic content in the extract prepared using 15 different extraction conditions showed good correlation with α-glucosidase inhibitory effects, which is consistent with our present study about α-glucosidase inhibitory xanthones.

Collectively, the extraction yield and efficacy of *C. formosum* ssp. *pruniflorum* vary depending on the extraction conditions, and an extract with excellent efficacy can be efficiently secured through optimization of the extraction conditions. In addition, consistent with the efficacy of the isolated components, which was demonstrated in this study, the phenolic compounds were important for the efficacy of this plant and can be used as reference components for future product development.

## 4. Conclusions

Comparison of the roots, stems and leaves of *C. formosum* ssp. *pruniflorum* showed differences in the chemical profiles and biological activity. An investigation of *C. formosum* ssp. *pruniflorum* led to the isolation of 25 phenolic compounds, including 2 new compounds. The structures of the isolated compounds were determined to be xanthones, benzophenone, flavonoids and phenol. Two new compounds were defined as pruniflonone A (**16**) and pruniflonone B (**19**). The isolated compounds showed good antioxidant and α-glucosidase inhibitory activity with differences in activity depending on the structures. Optimization of extraction conditions was also studied using RSM for maximum efficacy. In conclusion, the *C. formosum* ssp. *pruniflorum* with antioxidant and α-glucosidase inhibitory activity might be beneficial for glucose-related diseases.

## Figures and Tables

**Figure 1 antioxidants-12-00511-f001:**
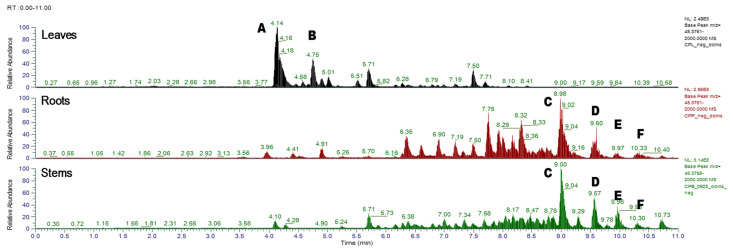
HRESI-MS/MS chromatograms of leaves, roots and stems of *C. formosum* ssp. *pruniflorum*.

**Figure 2 antioxidants-12-00511-f002:**
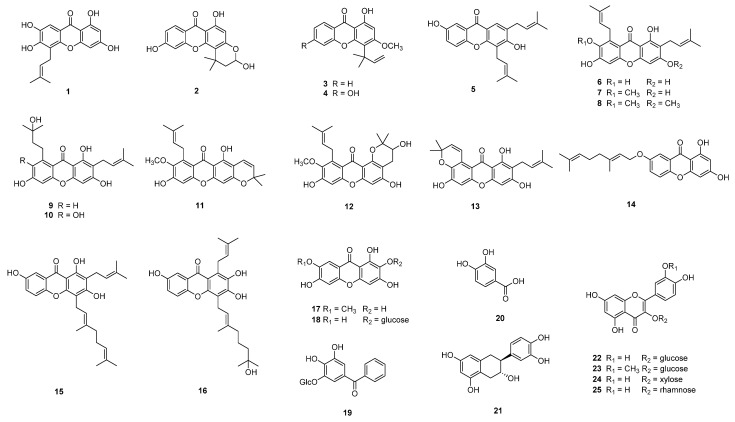
Chemical structures of compounds **1**–**25** from *C. formosum* ssp. *pruniflorum*.

**Figure 3 antioxidants-12-00511-f003:**
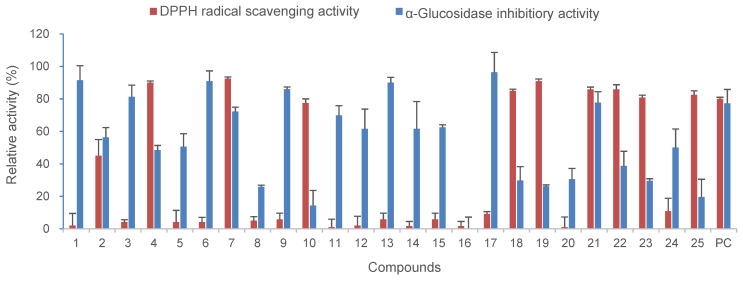
Antioxidant and α-glucosidase inhibitory activity of compounds **1**–**25** from *C. formosum* ssp. *pruniflorum*.

**Figure 4 antioxidants-12-00511-f004:**
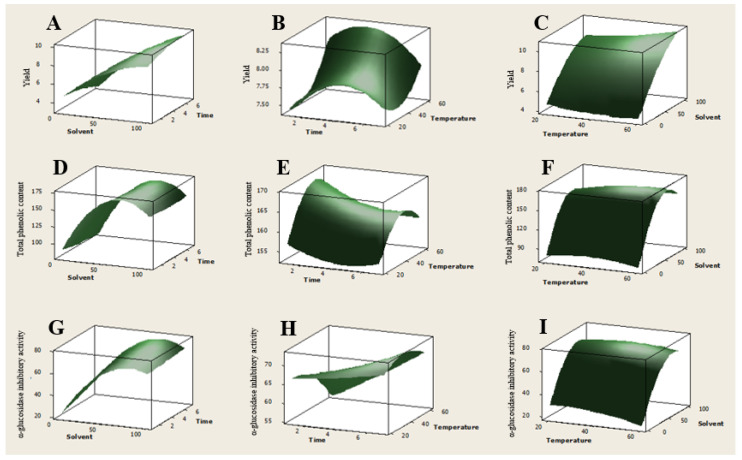
Response surface plots. Effect of extraction variables in yield (**A**–**C**), total phenolic content (**D**–**F**) and α-glucosidase inhibitory activity (**G**–**I**). Three variables are extraction solvent (X_1_), extraction time (X_2_) and extraction temperature (X_3_).

**Table 1 antioxidants-12-00511-t001:** Total phenolic and flavonoid content of leaves, roots and stems of *C. formosum* ssp. *pruniflorum*.

	Antioxidant Activity (IC_50_, μg/mL)	α-Glucosidase Inhibition (IC_50_, μg/mL)	Total Phenolic Content (mg GAE/g Extracts)	Total Flavonoid Content (mg CE/g Extracts)
Leaves	14.9	3.9	132.3	101.6
Roots	17.0	2.0	94.9	49.8
Stems	47.8	23.0	85.5	28.7

**Table 2 antioxidants-12-00511-t002:** Peak profiling by LC-MS/MS from leaves, roots and stems of *C. formosum* ssp. *pruniflorum*.

Peak No.	Compound Identification	t_R_ (min)	m/z		Molecular Formular [M-H]^−^	UV (λ_max_, nm)	Compd No in This Study
			Observed	Calculated			
**A**	mangiferin	4.14	421.0769	421.0776	C_19_H_18_O_11_	204, 256, 316, 364	**18**
**B**	quercetin-3-*O*-glucopyranoside	4.75	463.0877	463.0882	C_21_H_20_O_12_	204, 256, 356	**22**
**C**	γ-mangostin	9.00	395.1507	395.1500	C_23_H_23_O_6_	208, 268, 316, 364	**12**
**D**	α-mangostin	9.57	409.1664	409.1657	C_24_H_26_O_6_	240, 316	**8**
**E**	7-geranyloxy-1,3-dihydroxyxanthone	9.96	379.1541	379.1551	C_23_H_24_O_5_	224, 236, 260, 308, 368	**1**
**F**	cochinchinone A	10.31	447.2177	447.2177	C_28_H_32_O_5_	220, 240, 268, 316, 408	**5**

**Table 3 antioxidants-12-00511-t003:** A Box–Behnken Design for independent variables and their responses.

Run	Actual Variables (Coded Variables)			Observed Values		
	Extraction Solvent (X_1_, %)	Extraction Time (X_2_, h)	Extraction Temperature (X_3_, °C)	α-Glucosidase Inhibition (% of Control)	Yield (%)	Total Phenolic Content (mg GAE/g Extract)
1	100 (1)	7 (1)	40 (0)	67.9	10.30	154.2
2	0 (−1)	4 (0)	20 (−1)	29.2	4.76	80.2
3	100 (1)	4 (0)	60 (1)	67.2	10.58	153.9
4	50 (0)	1 (−1)	20 (−1)	64.3	7.73	159.4
5	50 (0)	4 (0)	40 (0)	70.7	7.70	165.4
6	50 (0)	1 (−1)	60 (1)	53.5	8.41	166.1
7	0 (−1)	7 (1)	40 (0)	34.6	3.50	82.1
8	50 (0)	4 (0)	40 (0)	68.5	7.74	165.1
9	100 (1)	1 (−1)	40 (0)	68.3	8.68	153.3
10	50 (0)	7 (1)	20 (−1)	72.2	7.31	154.9
11	100 (1)	4 (0)	20 (−1)	68.1	9.21	130.5
12	0 (−1)	1 (−1)	40 (0)	25.3	4.04	87.3
13	50 (0)	7 (1)	60 (1)	71,8	7.59	157.8
14	50 (0)	4 (0)	40 (0)	64.3	8.01	159.3
15	0 (−1)	4 (0)	60 (1)	18.6	4.31	78.2

**Table 4 antioxidants-12-00511-t004:** ANOVA for response surface regression equation.

Responses	Category	Sum of Square	Degree of Freedom	Mean Square	*F* Value	*p* Value
Yield	Regression	67.3248	9	7.4805	45.57	<0.001
	Linear	61.7689	3	20.5896	125.44	<0.001
	Square	3.5177	3	1.1726	7.14	0.029
	Interaction	2.0382	3	0.6794	4.14	0.08
	Residual error	0.8207	5	0.1641		
	Lack-of-fit	0.7439	3	0.248	6.45	0.137
	Pure error	0.0769	2	0.0384		
	Total	68.1455	14			
	R^2^ = 0.988, adjusted R^2^ = 0.966					
Total phenolic	Regression	17,170.61	9	1907.85	139.74	<0.001
	Linear	8868.86	3	2956.29	216.53	<0.001
	Square	8126.41	3	2708.8	198.4	<0.001
	Interaction	178.34	3	58.45	4.28	0.076
	Residual error	68.27	5	13.65		
	Lack-of-fit	42.03	3	15.01	1.29	0.464
	Pure error	23.24	2	11.62		
	Total	17,238.9	14			
	R^2^ = 0.996, adjusted R^2^ = 0.989					
α-Glucosidase	Regression	5033.5	9	559.28	37.13	<0.001
inhibition	Linear	9570.43	3	1190.14	79.02	<0.001
	Square	1389.17	3	463.06	30.74	0.001
	Interaction	73.9	3	24.63	1.64	0.294
	Residual error	75.31	5	15.06		
	Lack-of-fit	53.81	3	17.94	1.67	0.396
	Pure error	21.5	2	10.75		
	Total	5108.81	14			
	R^2^ = 0.985, adjusted R^2^ = 0.959					

**Table 5 antioxidants-12-00511-t005:** Predicted and observed values of yield, total phenolic content and α-glucosidase inhibitory activity under optimized condition.

Optimized Extraction Condition			Responses					
Extraction Solvent (% MeOH in EtOAc)	Extraction Time (h)	Extraction Temperature (°C)	α-Glucosidase Inhibitory Activity ^a^ (% of Control)		Yield (%)		Total Phenolic Content (mg GAE/g extract)	
88.1	6.02	60.0	Predicted	Observed	Predicted	Observed	Predicted	Observed
			72.2	73.9	10.4	10.9	163.9	163.9

^a^ α-Glucosidase inhibitory activity (%) was measured at 1 μg/mL.

## Data Availability

The data are contained within the article or Appendix A.

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
