# Peer review of "Characterization of Antioxidant and α-Glucosidase Inhibitory Compounds of Cratoxylum formosum ssp. pruniflorum and Optimization of Extraction Condition"

_antioxidants, 2023, doi:10.3390/antiox12020511_

Round 1

Reviewer 1 Report

The article "Characterization of antioxidant and α-glucosidase inhibitory compounds of Cratoxylum formosum ssp. pruniflorum and optimization of extraction condition" presents data regarding the antioxidant effect and inhibitory effect on Glucosidase, presenting also the chemical composition of the extracts obtained in methanol from the main componennts of this plant (roots, leaves and stems). The novelty of the work is only in terms of the two new compounds identified in the extract: pruniflonone A (16) and pruniflonone B (19), as chemical composition of this plant has been extensively studied (https://doi.org/10.1016/j.fitote.2014.02.002, 10.1055/s-0041-111621; https://doi.org/10.1016/j.bmcl.2017.07.066, DOI:10.7314/APJCP.2015.16.14.6117, DOI: 10.2174/1570178616666190902111630, https://doi.org/10.1021/np1001797).

Introduction: The introduction must be completed with data regarding more information about the plant (Cratoxylum formosum ssp. pruniflorum). Please present briefly previous data obtained by other researchers regarding the chemical composition of C. formosus, and their roles in human health by acting as antioxidant and anticancer agents (biological assessments). Please clearly state the novelty of this article.

Experimental The extraction is basically done. Several other methods could be applied for this step: ultrasound assisted extraction, etc. Why have you chosen methanol as solvent? More information on the successively partition with CH2Cl2, EtOAc and n-BuOH could be added.

Results: What was the experimental extraction yield obtained? The results should be discussed in comparation with results obtained by other researchers (10.1016/j.fitote.2014.02.002, https://doi.org/10.1021/np1001797, https://doi.org/10.1016/j.fitote.2014.02.002 etc) in terms of extraction yield and antioxidant activities.

Please explain further: "Related to α-glucosidase activity, only compound 21 without any sugar moieties showed inhibition." In Fig. 3, this is not very obvious - several compounds have quite high % of -Glucosidase inhibitory activity. Was the extract analyzed as glucosidase inhibitor or each compound separately? Please add discussion related to the newly discovered compounds and their effect (antioxidative and inhibitory).

In the optimization part- the obtained yield is very low - 10.9%. How do you explain this? Is it effective from an economic point of view to work with so low yields? Please as some details related to column 1 - (% MeOH in EtOAc) as extraction solvent, as in the experimental part only MeOH was presented for extraction.

Author Response

The article "Characterization of antioxidant and α-glucosidase inhibitory compounds of Cratoxylum formosum ssp. pruniflorum and optimization of extraction condition" presents data regarding the antioxidant effect and inhibitory effect on Glucosidase, presenting also the chemical composition of the extracts obtained in methanol from the main componennts of this plant (roots, leaves and stems). The novelty of the work is only in terms of the two new compounds identified in the extract: pruniflonone A (16) and pruniflonone B (19), as chemical composition of this plant has been extensively studied (https://doi.org/10.1016/j.fitote.2014.02.002, 10.1055/s-0041-111621; https://doi.org/10.1016/j.bmcl.2017.07.066, DOI:10.7314/APJCP.2015.16.14.6117, DOI: 10.2174/1570178616666190902111630, https://doi.org/10.1021/np1001797).

Introduction: The introduction must be completed with data regarding more information about the plant (Cratoxylum formosum ssp. pruniflorum). Please present briefly previous data obtained by other researchers regarding the chemical composition of C. formosus, and their roles in human health by acting as antioxidant and anticancer agents (biological assessments). Please clearly state the novelty of this article.

[Answer] As suggested, previous data about chemical composition and biological were added. The novelty of this articles was also described. Thank you.

Experimental: The extraction is basically done. Several other methods could be applied for this step: ultrasound assisted extraction, etc. Why have you chosen methanol as solvent? More information on the successively partition with CH2Cl2, EtOAc and n-BuOH could be added.

[Answer] Methanol was chosen as a solvent for extraction due to its broad solubility for constituents. More information about the extraction and fractionation procedure were added, as suggested. Thank you.

Results: What was the experimental extraction yield obtained? The results should be discussed in comparation with results obtained by other researchers (10.1016/j.fitote.2014.02.002, https://doi.org/10.1021/np1001797, https://doi.org/10.1016/j.fitote.2014.02.002 etc) in terms of extraction yield and antioxidant activities.

[Answer] As suggested, the extraction yield was compared with previous studies. Although the extraction solvents were not the same, they also used mixture of alcohol and water and showed similar yield with our study. To the best of our knowledge, the antioxidant activity of the constituents of this plant has not been reported yet. Thank you for your careful review.

Please explain further: "Related to α-glucosidase activity, only compound 21 without any sugar moieties showed inhibition." In Fig. 3, this is not very obvious - several compounds have quite high % of -Glucosidase inhibitory activity. Was the extract analyzed as glucosidase inhibitor or each compound separately? Please add discussion related to the newly discovered compounds and their effect (antioxidative and inhibitory).

[Answer] To clarify, the manuscript was revised as “Related to α-glucosidase activity of flavonoids of C. formosum ssp. pruniflorum”. The activities were tested for both extract and isolated compounds, which were presented as Table 1 and Figure 3, respectively. Discussion about the newly discovered compounds was added, as suggested.

In the optimization part- the obtained yield is very low - 10.9%. How do you explain this? Is it effective from an economic point of view to work with so low yields? Please as some details related to column 1 - (% MeOH in EtOAc) as extraction solvent, as in the experimental part only MeOH was presented for extraction.

[Answer] First, the samples were extracted with methanol, and bioactive compounds were purified and identified. Based on that results, optimization of the conditions for the content of the bioactive compounds was performed. The solvent conditions were selected as the mixture of EtOAc and MeOH in a preliminary study and optimization was carried out through RSM.

Reviewer 2 Report

Manuscript antioxidants-2213921, is an innovative experimental work, well designed performed but it needs a better presentation.

There is virtually no Discussion (as also shown in title: 3. Results) in this manuscript! The references used are numbered in one hand and the relevant discussion is very general. There should be more specific discussion at least on the antioxidant activity/phenolics and RSM.

Further, I had a difficulty in reading it since the order of presentation in the M&M is different than the that of the R or the abstract.

In M&M the order of presentation is:

the plant material (here the leaves, roots and stems should be described better, e.g. mature leaves?),

the General Experimental Procedure (this refers rather to Instumentation)

the Analysis of Chemical Profile using LC-MS/MS (methodology)

the Extraction and Isolation (partitioning with CH2Cl2, EtOAc and n-BuOH, however, results are given only with CH2Cl2, EtOAc)

in the of Pruniflonone A (16) & Pruniflonone B (19) spectral conditions/characteristics are given without any others explanation of what it is.

Then continues with :

2.5. Measurement of Antioxidant and α-Glucosidase Activity

2.5 (should be 2.6.) Quantitation of Phenolic and Flavonoid Contents

2.6. (Should be 2.7) Response Surface Methodology

Whereas in the R the order is inverse:

- Comparison of different parts of C. formosum ssp. pruniflorum

-Isolation and Characterization of the constituents of C. formosum ssp. pruniflorum

-Structure Elucidation of New Compounds

-Identification of Known Compounds

-Evaluation of Antioxidant and α-Glucosidase Inhibitory Activity

-Optimization of extraction conditions using response surface metholodogy

Additionally, in the abstract is stated that <<Further analysis showed the positive correlation between a-glucosidase inhibitory activity and total phenolic content>>. It is not clear in the R where this statement is coming from.

In 2.4., 2nd paragraph,  line 5: <<trile-H2O, 80:20) of CPRC5 and CPRC17 gives compounds 10 and 14, and compounds 4>>, SHOULD in the past (gave).

To my opinion, the article does provide with innovative information, however, it should be present in a more comprehensive order and the results should be discussed appropriately.

Author Response

Manuscript antioxidants-2213921, is an innovative experimental work, well designed performed but it needs a better presentation.

There is virtually no Discussion (as also shown in title: 3. Results) in this manuscript! The references used are numbered in one hand and the relevant discussion is very general. There should be more specific discussion at least on the antioxidant activity/phenolics and RSM.

Further, I had a difficulty in reading it since the order of presentation in the M&M is different than the that of the R or the abstract.

 [Answer] As suggested, the order of Experimental Section was changed to unify the order of presentation.

In M&M the order of presentation is:

the plant material (here the leaves, roots and stems should be described better, e.g. mature leaves?),

[Answer] Mature parts were collected and the description of plant material was added to Experimental Section.

the General Experimental Procedure (this refers rather to Instumentation)

the Analysis of Chemical Profile using LC-MS/MS (methodology)

the Extraction and Isolation (partitioning with CH2Cl2, EtOAc and n-BuOH, however, results are given only with CH2Cl2, EtOAc)

 [Answer] Based on our preliminary data, CH2Cl2 and EtOAc-soluble fractions with good activity and characteristic components were selected for further purification.

in the of Pruniflonone A (16) & Pruniflonone B (19) spectral conditions/characteristics are given without any others explanation of what it is.

[Answer] The structures were determined on the basis of spectral data in the Results and Discussion. Thank you.

Then continues with :

2.5. Measurement of Antioxidant and α-Glucosidase Activity

2.5 (should be 2.6.) Quantitation of Phenolic and Flavonoid Contents

2.6. (Should be 2.7) Response Surface Methodology

Whereas in the R the order is inverse:

- Comparison of different parts of C. formosum ssp. pruniflorum

-Isolation and Characterization of the constituents of C. formosum ssp. pruniflorum

-Structure Elucidation of New Compounds

-Identification of Known Compounds

-Evaluation of Antioxidant and α-Glucosidase Inhibitory Activity

-Optimization of extraction conditions using response surface metholodogy

 [Answer] As suggested, the order of the experimental s was changed to unify the order of presentation.

Additionally, in the abstract is stated that <<Further analysis showed the positive correlation between a-glucosidase inhibitory activity and total phenolic content>>. It is not clear in the R where this statement is coming from.

 [Answer] As pointed out, the sentence was deleted.

In 2.4., 2nd paragraph,  line 5: <<trile-H2O, 80:20) of CPRC5 and CPRC17 gives compounds 10 and 14, and compounds 4>>, SHOULD in the past (gave).

[Answer] The error was corrected, as pointed out.

To my opinion, the article does provide with innovative information, however, it should be present in a more comprehensive order and the results should be discussed appropriately.

[Answer] As suggested, the order of presentation was adjusted to make it easier to be understood. In addition, the meaning of the results of this study was suggested by adding discussions on the efficacy of the isolated compound and the differences between parts. Thank you so much for your careful review.

Reviewer 3 Report

Dear authors the work is interesting although the approach to my opinion will match more to a natural product research journal rather than to antioxidants

Please provide more details or appropriate reference for total flavonoid content as ref.20 does not give such a description but only for total phemol. Seeing the results in Table 1, that is the expression in catechin equivalents, I deduce that expect for AlCl3, probably NaNO2 has been used in the protocol. If this is the case, to my knowledge, not only flavonoids will react.

Considering the optimization part, why the authors used  instead of Central Composite Design the less robust Box-Behnken which uses just three levels of each factor instead of 5, is nearly, thus, rotatable and cannot be applied to  up to a fourth-order model.

Finally, it is not clear to me when the authors consider the solvent, how they came up with the decission of a mixture of methanol and ethylacetate

Author Response

The work is interesting although the approach to my opinion will match more to a natural product research journal rather than to antioxidants

[Answer] Our present study showed the antioxidant activity of the extract of different part of C. formosum ssp. pruniflorum. In addition, the antioxidant activity of isolated compounds was also evaluated. Oxidative stress is closely related to metabolic diseases including diabetes. Therefore, it will provide useful information about the antioxidant potential and its application of C. formosum ssp. pruniflorum and its compounds.

Please provide more details or appropriate reference for total flavonoid content as ref.20 does not give such a description but only for total phenol. Seeing the results in Table 1, that is the expression in catechin equivalents, I deduce that expect for AlCl3, probably NaNO2 has been used in the protocol. If this is the case, to my knowledge, not only flavonoids will react.

[Answer] As pointed out, reference for total flavonoid was added. Thank you for your careful review.

Considering the optimization part, why the authors used instead of Central Composite Design the less robust Box-Behnken which uses just three levels of each factor instead of 5, is nearly, thus, rotatable and cannot be applied to up to a fourth-order model. Finally, it is not clear to me when the authors consider the solvent, how they came up with the decussion of a mixture of methanol and ethylacetate.

[Answer] The ranges of each variable such as extraction solvent, extraction temperature, and extraction time were determined on the preliminary study. We supposed that three factor-three level was appropriate for this purpose. Discussion about the RSM was added, as suggested. Thank you for your careful review.

Round 2

Reviewer 2 Report

The authors responded to the commends to an acceptable degree and thus the article can be accepted for publication